# Discharge and Temperature Controls of Dissolved Organic Matter (DOM) in a Forested Coastal Plain Stream

Yuehan Lu [1,2,*], Peng Shang [1], Shuo Chen [1], Yingxun Du [3], Marco Bonizzoni [2,4] and Amelia K. Ward [5]

1 Molecular Eco-Geochemistry (MEG) Laboratory, Department of Geological Sciences, University of Alabama, Tuscaloosa, AL 35487, USA; pshang1@crimson.ua.edu (P.S.); schen83@crimson.ua.edu (S.C.)
2 Alabama Water Institute, University of Alabama, Tuscaloosa, AL 35487, USA; marco.bonizzoni@ua.edu
3 State Key Laboratory of Lake Science and Environment, Nanjing Institute of Geography and Limnology, Chinese Academy of Sciences, Nanjing 210008, China; yxdu@niglas.ac.cn
4 Department of Chemistry & Biochemistry, University of Alabama, Tuscaloosa, AL 35487, USA
5 Department of Biological Sciences, University of Alabama, Tuscaloosa, AL 35487, USA; award@ua.edu
* Correspondence: Yuehan.lu@ua.edu

**Abstract:** Streams in the southeastern United States Coastal Plains serve as an essential source of energy and nutrients for important estuarine ecosystems, and dissolved organic matter (DOM) exported from these streams can have profound impacts on the biogeochemical and ecological functions of fluvial networks. Here, we examined hydrological and temperature controls of DOM during low-flow periods from a forested stream located within the Coastal Plain physiographic region of Alabama, USA. We analyzed DOM via combining dissolved organic carbon (DOC) analysis, fluorescence excitation–emission matrix combined with parallel factor analysis (EEM-PARAFAC), and microbial degradation experiments. Four fluorescence components were identified: terrestrial humic-like DOM, microbial humic-like DOM, tyrosine-like DOM, and tryptophan-like DOM. Humic-like DOM accounted for ~70% of total fluorescence, and biodegradation experiments showed that it was less bioreactive than protein-like DOM that accounted for ~30% of total fluorescence. This observation indicates fluorescent DOM (FDOM) was controlled primarily by soil inputs and not substantially influenced by instream production and processing, suggesting that the bulk of FDOM in these streams is transported to downstream environments with limited in situ modification. Linear regression and redundancy analysis models identified that the seasonal variations in DOM were dictated primarily by hydrology and temperature. Overall, high discharge and shallow flow paths led to the enrichment of less-degraded DOM with higher percentages of microbial humic-like and tyrosine-like compounds, whereas high temperatures favored the accumulation of high-aromaticity, high-molecular-weight, terrestrial, humic-like compounds in stream water. The flux of DOC and four fluorescence components was driven primarily by water discharge. Thus, the instantaneous exports of both refractory humic-like DOM and reactive protein-like DOM were higher in wetter seasons (winter and spring). As high temperatures and severe precipitation are projected to become more prominent in the southeastern U.S. due to climate change, our findings have important implications for future changes in the amount, source, and composition of DOM in Coastal Plain streams and the associated impacts on downstream carbon and nutrient supplies and water quality.

**Keywords:** DOM; fluorescence; Coastal Plain stream; carbon export; flow path; climate change; biodegradation; EEM-PARAFAC; redundancy analysis

## 1. Introduction

Dissolved organic matter (DOM) is a complex mixture of organic compounds with various compositions, molecular weights, and reactivities. DOM in aquatic environments is broadly classified as allochthonous, which refers to compounds originating from soils and decayed terrestrial plants, or autochthonous, which refers to those from instream organisms





(macrophytes and microbes, including microalgae). Because of the source and compositional complexity, DOM plays multifaceted roles in regulating surface water quality. For example, DOM can increase the solubility and mobility of metals and organic pollutants and thus alter their bioavailability [1,2] and can facilitate the formation of carcinogenic disinfection byproducts (DBPs) during the chlorination of drinking water [3,4]. Additionally, DOM serves as a source of N and P nutrients and fuels microbial respiration, thereby contributing to dissolved oxygen depletion and the creation of the seasonally hypoxic "dead zones" in coastal oceans [5]. Furthermore, DOM contributes to light absorption, which provides protection to aquatic biota from harmful radiation but at the same time can also limit photosynthesis [6]. The various environmental and ecological roles that natural DOM plays, however, depend on its source and composition, which are tightly coupled with a variety of environmental drivers [7–9]. Identifying the link between DOM source–composition variability and associated environmental drivers is the first necessary step toward integrating DOM in water quality regulation and management.

Small watershed streams represent the primary interface where terrestrial materials, including DOM, enter aquatic environments [10,11]. Forested Coastal Plain streams usually exhibit high rates of organic carbon export and are considered hotspots of exporting organic substrates to downstream environments [12,13]. These streams typically carry a large quantity of dissolved organic carbon (DOC) and colored DOM (CDOM) due to usually abundant rainfalls and fertile forest soils that contain a significant litter layer (O horizon) enriched with organic substances from leaf litter, roots, microbes, and fungi. As the organic substances percolate downward in soils, they are altered by physical sorption and microbial processing, leading to quantity and quality variations in soil OM [14,15]. Overall, DOM leached from deeper soils has lower concentrations but is relatively enriched in low-molecular-weight, low-aromaticity compounds that are less preferentially sorbed to soil minerals [14]. As a result, the soil-to-stream hydrological flow path that often varies with precipitation and water table exerts an important control on the amount, source, and composition of DOM exported from small, forested watersheds [16,17]. Furthermore, temperature and moisture influence the production and decomposition of organic materials in both soils and streams, and consequently, DOM exported by small, forested streams.

Climate change can lead to significant changes in hydrology and temperature and hence DOM exported across the terrestrial–aquatic interface. Identifying hydroclimatic drivers for the quantity and quality of DOM from small streams is of increasing importance in a rapidly changing climate. Although this topic has been a subject of much research [18–21], few have focused on Coastal Plain streams in the southeastern United States, where climate change is projected to enhance the seasonality of temperature and shift the pattern of precipitation across the entire region [22]. Mean annual temperatures in the Southeast have increased by about 2 °F since 1970 and are projected to increase by 4–8 °F by 2100, and seasonal contrast is predicted to increase, as the greatest warming is expected to occur during summer [22]. Precipitation patterns are also projected to change, with different effects on different areas (some areas become drier, while others become wetter), but substantial increases in extreme storm events are expected due to Atlantic hurricane activity. Identifying hydrological and temperature controls of DOM in southeastern U.S. streams is thus crucial to predict how these ecologically important streams will respond in the face of climate change.

Here, we examined temporal variations in DOM from a forested stream located within the Coastal Plain physiographic region of Alabama, southeastern U.S. We analyzed DOM via combining dissolved organic carbon (DOC) analysis, fluorescence excitation–emission matrix combined with parallel factor analysis (EEM-PARAFAC), and microbial degradation experiments. The objectives of our study were to: (i) determine the seasonal pattern of amount, source, and composition of Coastal Plain stream water DOM; and (ii) determine what effect changes in hydroclimatic factors have on Coastal Plain stream DOM, especially the two factors immediately influenced by climate changes—temperature and discharge. This research adds to the few studies that have characterized DOM in Coastal Plain streams

in the southeastern United States (e.g., [12]). Our results provide insights into potential shifts in energy and substrates exported from small, forested Coastal Plain watersheds and associated impacts on water quality and fluvial biogeochemical functions due to climate change.

## 2. Materials and Methods

### 2.1. Study Site

The study site, Mayfield Creek, is a second-order (Strahler scale) stream located in the Oakmulgee District of the Talladega National Forest in west-central Alabama, southeastern United States (Figure 1). The creek flows south to north and is a tributary to streams that drain into the Black Warrior River, which is an important inland waterway connecting the northern and southern parts of Alabama to transport coal, petroleum, and other goods. As part of headwater networks of the Mobile River Basin, Mayfield Creek drains a 17.5 km$^2$ watershed with a gentle watershed slope (0.2° on average). The watershed is dominated by a temperate, mixed deciduous forest (>98% of land use) comprised of pine (e.g., longleaf, shortleaf, yellow, and loblolly) and hardwood (e.g., oak, hickory, sweetgum, dogwood). The entire watershed is underlain by Cretaceous unconsolidated sands, and streambed sediments are composed of fine sand and gravelly sand. The average depth of the soil layer in the study catchment is greater than 2 m. The surface soil is mainly dark yellowish-brown sandy and flaggy loam, and the subsoil is yellowish red sandy and clay loam, according to Natural Resources Conservation Service Soils Database. The study area is influenced by a humid, subtropical climate with hot, humid summers and mild, wet winters. During the study period from 2014 to 2016, the mean annual temperature was 15 °C, and the mean annual precipitation was 1300 mm.

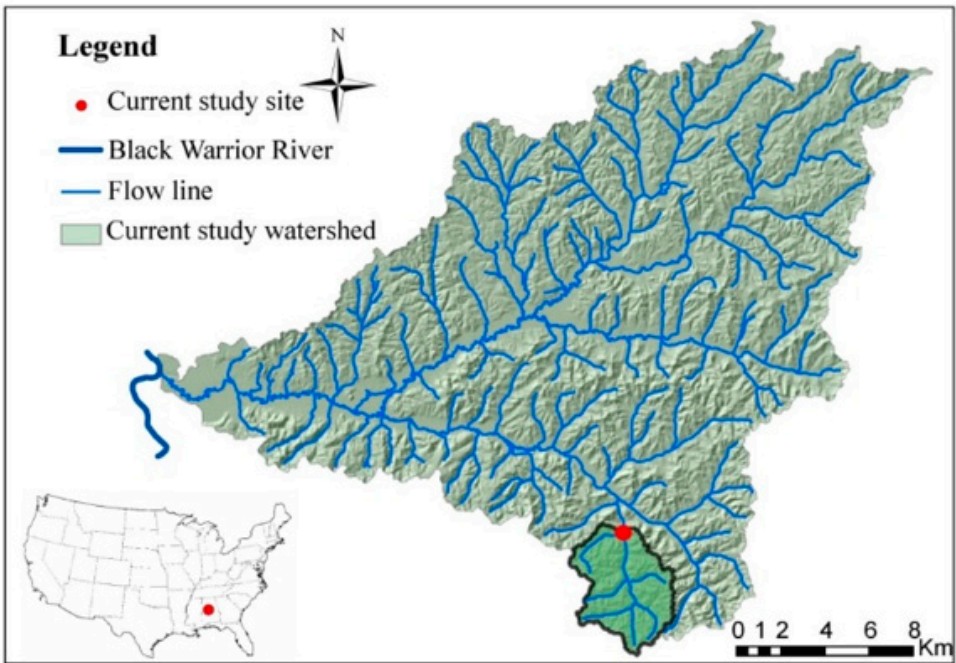

**Figure 1.** A map of the study area and sampling site. Streams are indicated by blue lines. The watershed boundary of the Mayfield Creek is indicated by black lines, and the sampling site is denoted by a red dot.

### 2.2. Flow Characterization and Sample Collection

Stream water level data were recorded continuously by a HOBO pressure transducer and data logger every 10 to 15 min during the study period (Supplementary Figure S1). To convert the logged water level data to stream discharge, a rating curve was established (Supplementary Figure S2). The stream water discharge data used in the rating curve

were measured following the velocity-area method from United States Geological Survey (USGS). Briefly, a uniform section of the stream was selected to measure the depth using a ruler and the velocity (m s$^{-1}$) every 50 cm using a portable flowmeter (Marsh-McBirney, Inc., Frederick, MD, USA, model 2000), and water discharge was calculated by multiplying the area by the mean velocity of each subsection and then summing across the subsections.

To evaluate seasonal variations in DOM from Mayfield Creek, a total of 35 samples were collected every 10 days on average from 13 September 2015 to 27 July 2016 (Supplemental Tables S1 and S2). Water samples were collected near the water-level logging station (Supplemental Figure S2), and this sampling location shares similar physical characteristics with other accessible reaches of this stream, including well-developed riparian canopy, a brownish water color, and sandy streambed sediments. To collect samples for stable oxygen and hydrogen isotopic analyses ($\delta^2$H and $\delta^{18}$O), sampling bottles were fully immersed underwater to fill the bottle without air bubbles and headspace. These samples were stored in the dark under room temperature prior to analysis. For other geochemical measurements, including DOM, water samples were filtered through 0.2 μm VWR syringe polyethersulfone filters on the same day of sample collection, and filtrates were collected. The filtrates for the analysis of DOC and nutrient concentrations were stored at $-20$ °C in the dark, and samples for DOM optical measurements were preserved at 4 °C in the dark and analyzed within two weeks to avoid any potential interference from freezing–thawing processes [23,24]. The samples for cation analysis were acidified with ultrapure concentrated HNO$_3$ (2% by volume) and stored in a refrigerator.

### 2.3. DOC Concentration and DOM Optical Property

The measurements of DOC and DOM optical property followed the method described in detail in [7] and [25]. Dissolved organic carbon concentration was analyzed on a Shimadzu TOC-V total organic carbon analyzer, and the relative standard deviation calculated from duplicate measurements ranged between 0% and 3%. Ultraviolet-visible absorbance was collected using a Shimadzu UV-1800 spectrophotometer from the wavelengths of 190 to 670 nm at a 1 nm interval. DOM fluorescence was measured using a Horiba FluoroMax3 spectrofluorometer with the excitation wavelengths ranging from 240 to 500 nm every 5 nm and emission wavelengths ranging from 280 to 538 nm every 3 nm. The spectra were corrected for blanks, the inner filter effect, and the manufacturer's correction factors before the normalization relative to the area under the water Raman peak at the excitation wavelength of 350 nm [26]. DrEEM toolbox in MATLAB was used to acquire and validate fluorescence components [27]. The EEM model was generated from 270 samples, and it was validated via the split-half analysis by randomly assigning the samples into four quarter splits and performing three validation tests on the six combined dataset halves (S4C6T3) [27]. The abundance of each identified fluorescence component (Ci) was reported as the absolute amount (Fmax, R.U.) or percentage contributions of total fluorescence (%Ci = Fmax of Ci/total Fmax).

A suite of DOM source-composition indices was calculated from the absorption and fluorescence spectra, including specific U.V. absorbance (SUVA$_{254}$), spectral slope ratio (S$_R$), fluorescence index (FI), humification index (HIX), and biological index (BIX). SUVA$_{254}$ (specific U.V. absorbance; L mg$^{-1}$ m$^{-1}$) was calculated by dividing the U.V. decadal absorption coefficient at 254 nm by DOC concentration [28]. S$_R$ was the ratio of spectral slope gradient of 275–295 nm to that of 350–400 nm, and spectral slope gradient was the slope of the natural logarithm of absorption coefficients over the corresponding wavelengths [29,30]. FI was the ratio of emission intensity at 470 nm to that at 520 nm at the excitation wavelength of 370 nm [26]. HIX was defined as the ratio of the integrated area under the emission wavelength of 435–480 nm to the sum of the area under 300–345 nm and the area under 435–480 nm when the excitation wavelength was 254 nm [31]. BIX was calculated as the ratio of emission intensity at 380 nm divided by 430 nm at the excitation of 310 nm [32].

### 2.4. Laboratory Incubations to Evaluate DOC Bioreactivity

Three additional sets of samples were collected on 1 October 2015, 26 April 2016, and 29 November 2016 for estimating DOM bioreactivity. The bioreactivity assay followed the method described in detail in ref. [25]. Stream waters were filtered first through precombusted 0.7 μm pore size GF/F filters and then through pre-cleaned 0.2 μm pore size filters (Whatman polycap) to remove bacteria. The filtrates were then inoculated by in situ microbes by adding 1% (by volume) unfiltered stream water from Mayfield Creek [33]. Each sample was distributed in three pre-combusted, one-liter amber glass bottles and incubated at 20 °C in the dark for 28 days. Over the course of the incubations, bottles were opened and shaken gently every day to ensure no oxygen exhaustion. Subsamples were collected on day 0, 5, and 28 and analyzed for DOC concentration and DOM optical properties (as described above). Percentage biodegradable DOM over five days (regarded as labile DOM) and 28 days (semi-labile DOM) was calculated as

$$\%\text{biodegradable DOM} = [(\text{DOM}_0 - \text{DOM}_t)/\text{DOM}_0] \times 100 \qquad (1)$$

where $\text{DOM}_0$ refers to DOC or fluorescence component intensity at day 0, and $\text{DOM}_t$ refers to DOC or fluorescence component intensity at day 5 or 28.

### 2.5. Ancillary Hydrological and Biogeochemical Parameters: Cations, Inorganic Nutrients, and Stable Oxygen and Hydrogen Isotopes of Water

The concentrations of cations were analyzed using a Perkin Elmer Optima 3000 DV ICP-OES, yielding the relative standard deviation for sample duplicate between 0 and 2%. Stable oxygen and hydrogen isotopic compositions of water were measured on a Picarro analyzer (WS-CRDS) at the Environmental and Natural Resources Institute (ENRI)-Stable Isotope Laboratory, University of Alaska. The relative standard deviation for the analysis of sample duplicate ranged between 0 and 3%. The concentrations of inorganic N and P nutrients (nitrate, ammonium, and phosphate) were measured using a Lachat QuikChem 8500 Flow Injection Ion Analyzer equipped with standard colorimetric modules designed for each analyte, following the QuikChem methods 10-107-04-1-B, 10-107-06-1-F, and 10-115-01-1-B. The relative standard deviation for samples collected in duplicate varied between 0 and 11%.

### 2.6. Statistical Analysis

All statistical analyses were conducted in the R statistical environment. The significance level, $\alpha$, was set at 0.05. First, we performed Pearson correlations between DOM indices ($\text{SUVA}_{254}$, $S_R$, HIX, FI, BIX) and percentages of Ci to constrain the source and nature of the identified fluorescence components. We also compared DOM fluxes across the seasons (spring vs. summer vs. autumn vs. winter) using the Kruskal–Wallis test and post hoc Dunn's pairwise multiple comparison test.

Furthermore, we identified environmental predictors for DOM source-compositional characteristics using generalized linear models (GLM) (R package 'MuMIm'). A suite of hydroclimatic variables was assessed as predictors, including stream discharge, water $\delta^2$H and $\delta^{18}$O, dissolved sodium concentration ($\text{Na}^+$), dissolved silica concentration (Si), temperature, nitrate, ammonium, and phosphate. To avoid multicollinearity among the predictors, the variance inflation factor (VIF) was calculated, and water $\delta^2$H was removed as a predictor to keep VIF < 8. The dependent variables included DOC and various DOM optical indices. Both explanatory and dependent variables were standardized prior to the modeling, and the model with the lowest AIC value was selected.

Finally, we applied redundancy analysis (RDA) to determine a linear combination of hydrological and biogeochemical predictors that best explained the matrix of DOM source-composition indices and identified sample grouping. The RDA model was fit by the rda() function of the R package 'vegan', and the significance of the model was evaluated by a permutation test (number of permutations: 999). The result was visualized with Type II scaling, i.e., correlation biplot. Both explanatory and dependent variables were standardized prior to the RDA.

## 3. Results

### 3.1. Physiochemical Parameters

In the study area, precipitation events were overall evenly distributed over four seasons, with the highest daily precipitation (7.37 cm) in late autumn. On the dates when samples were collected, daily rainfall ranged between 0 and 4.5 cm, and air and water temperatures varied from 1 to 33 °C and 7 to 27 °C, respectively. The stream water discharge of the entire sampling period ranged from 0.05 to 25.65 m$^3$/s, and the discharge on the sampling dates varied from 0.06 to 0.71 m$^3$/s, capturing low-flow periods of the year (Figure 2). The number of samples collected was 13 for autumn, 9 for winter, 8 for summer, and 5 for spring.

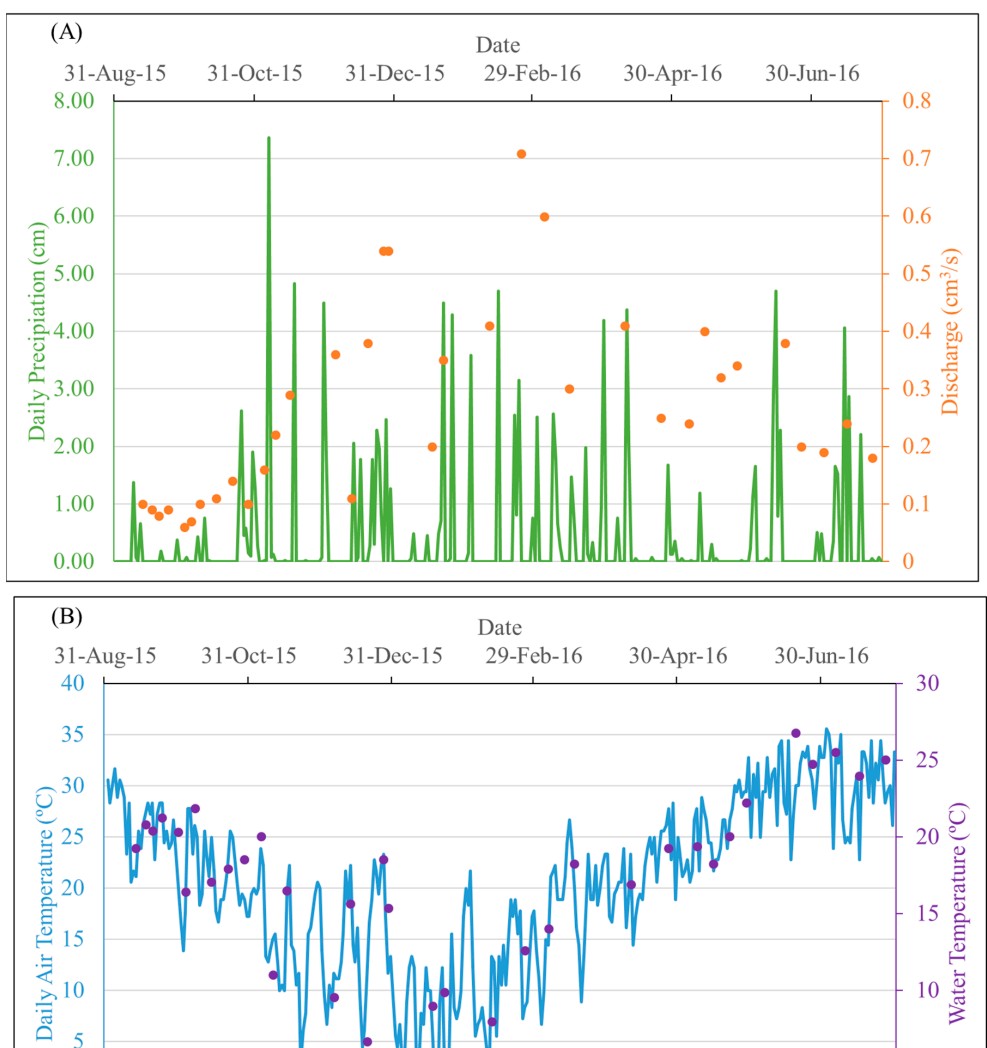

**Figure 2.** Temperature and hydrological variation of the study area over the study period. (**A**) Daily precipitation of the study area and discharge of the Mayfield Creek on sampling dates; (**B**) daily air temperature of the study area and water temperature of the Mayfield Creek on sampling dates. Daily precipitation and temperature data were acquired from NOAA monitoring station GHCND:USC00018380, which is ~34 km from the sampling site.

For inorganic N and P species, nitrate varied from 3.7 to 33.5 μg N/L, ammonium from below detection to 65.1 μg N/L, and phosphate from 1.0 to 4.1 μg P/L. Stable hydrogen ($δ^2$H) and oxygen ($δ^{18}$O) isotopic values of water fell in the range of −20.44‰ to 14.22‰

and −4.51‰ to 3.73‰, respectively, and they were strongly correlated (Pearson $r = 0.90$, $p < 0.001$, df = 32). Sodium concentration fluctuated from 0.71 to 1.38 mg/L, and dissolved silica varied from 3.10 to 4.84 mg/L.

### 3.2. DOC and DOM Optical Indices

Optical indices are commonly used to evaluate the source and composition of DOM. For absorbance-based indices, higher $SUVA_{254}$ values indicate greater contributions of aromatic compounds relative to aliphatic compounds [28,34], and $S_R$ decreases with increasing DOM molecular weights and increases with the degree of photodegradation [30,35]. In Mayfield Creek, $SUVA_{254}$ varied from 1.97 to 3.98 and averaged $2.77 \pm 0.54$ (L mg$^{-1}$ m$^{-1}$), and $S_R$ values were from 0.71 to 1.20 and averaged $0.80 \pm 0.09$. For fluorescence-based indices, a HIX value of 1–2 is typically associated with non-humified plant material, whereas fulvic acid extracts have HIX values >10 [31,36]. Low values of FI (~1.2) indicate the presence of humic-like substances that originate from vascular plants, and high values of FI (~1.8) are dominated by microbially derived organic matter [34]. BIX provides an estimate of the amount of recently produced and/or autochthonous DOM, with a higher value (>1) indicating the dominance of freshly produced DOM that has not been significantly altered by microbial or photochemical processing [8,37]. In Mayfield Creek, HIX, FI, and BIX were in the range of 1.7–16.05, 1.56–1.82, and 0.52–1.23, respectively, and they averaged $4.91 \pm 2.95$, $1.69 \pm 0.06$, and $0.90 \pm 0.17$, respectively. FI values suggest that Mayfield DOM was sourced from both terrestrial and microbial origins; BIX indicates a mixture of fresh and degraded compounds, and HIX suggests a wide range of humification degrees. Overall, DOM indices suggest that Mayfield DOM contained a mixture of compounds of different origins, aromaticity, molecular weights, and diagenetic status.

Four fluorescence components (C1–C4) of different source-compositional characteristics were identified (Figure 3). C1 and C2 accounted for $31.3 \pm 1.3\%$ and $38.0 \pm 1.1\%$ of the total fluorescence, respectively, and they were more abundant than C3 and C4, which comprised $16.0 \pm 0.9\%$ and $14.7 \pm 2.0\%$, respectively. Percentage contributions of C1 showed significant positive correlations with $SUVA_{254}$ and HIX but negative correlations with BIX and FI, whereas percentage C2 positively correlated with BIX and FI but negatively with HIX (Table 1). Percentage C3 was positively correlated with $S_R$, BIX, and FI but negatively correlated with $SUVA_{254}$ and HIX; percentage C4 did not exhibit significant correlations with any optical indices (Table 1).

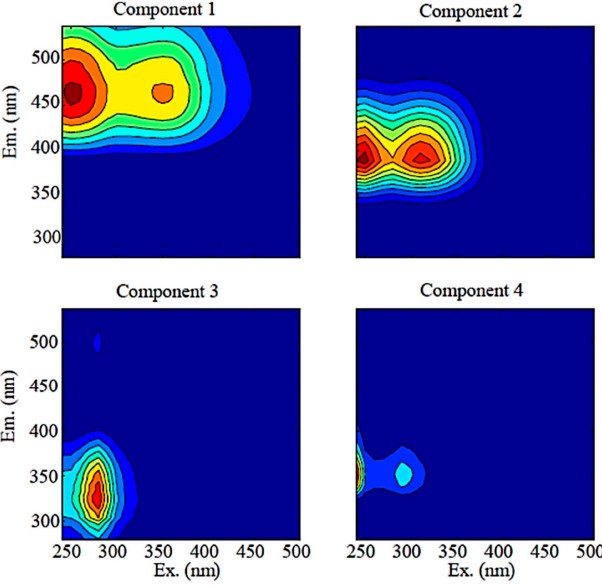

**Figure 3.** Excitation–emission contour plots of the four DOM fluorescence components (C1–C4) identified with DrEEM toolbox from the Mayfield Creek. Ex. = Excitation; Em. = Emission.

Percentage biodegradable DOC was in the range of 1.12–9.68% over 5 days and 3.04–22.66% over 28 days (Figure 4). Among the three sets of incubations, the 29 November 2016 sample had higher %BDOC (22.66 ± 2.17% over 28 days) than the samples from 1 October 2015 (3.04 ± 0.30%) and 26 April 2016 (6.91 ± 3.63%). Among the four fluorescence components, changes in C1 and C2 fluorescence were within ±20% for all samples over 5-day and 28-day incubations, which were much lower than percentage changes in C3 and C4. Both removal and production were observed for C3 and C4. C3 changed by −53% to +48% over 5 days and −286% to +64% over 28 days, and C4 changed by∞ to +32% over 5 days and ∞to +38% over 28 days (negative values indicate production and positive values indicate removal, and ∞ indicates that the initial value was 0).

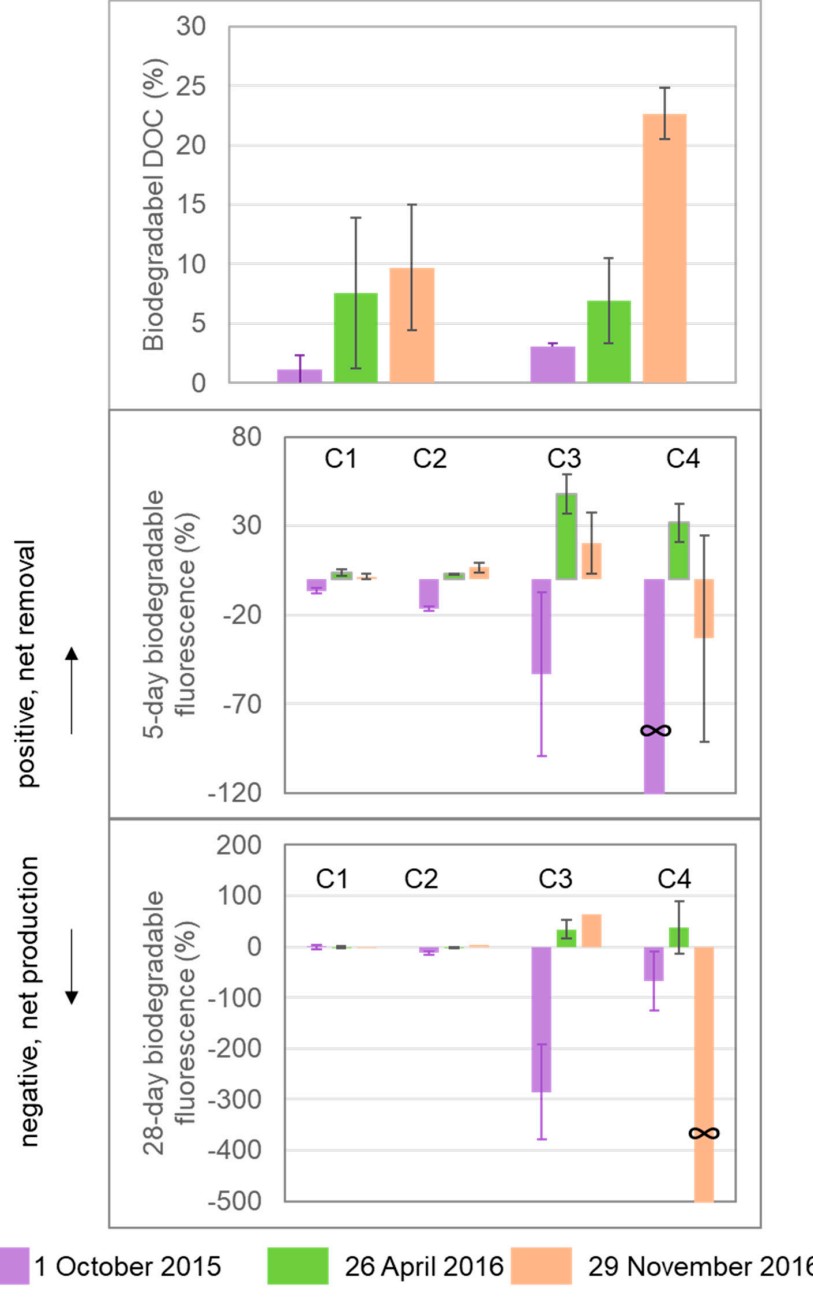

**Figure 4.** Percentage biodegradable DOC and DOM fluorescence components over the 5-day and 28-day laboratory incubations. Percentage biodegradable DOC or DOM was calculated as the difference between the initial and end values ($DOM_0 - DOM_t$) divided by the initial value, and "∞" indicates that the initial value was 0.

**Table 1.** Pearson correlation of DOM fluorescence components and optical indices in Mayfield Creek in the southeastern United States.

|  | SUVA$_{254}$ | HIX | S$_R$ | BIX | FI |
|---|---|---|---|---|---|
| %C1 [a] | **0.35, *p* = 0.04** | **0.76, *p* < 0.001** | −0.23, *p* = 0.17 | **−0.73, *p* < 0.001** | **−0.49, *p* < 0.001** |
| %C2 | −0.03, *p* = 0.87 | **−0.37, *p* = 0.03** | −0.13, *p* = 0.44 | **0.49, *p* < 0.001** | **0.56, *p* < 0.001** |
| %C3 | **−0.35, *p* = 0.04** | **−0.76, *p* < 0.001** | **0.53, *p* < 0.001** | **0.74, *p* < 0.001** | **0.67, *p* < 0.001** |
| %C4 | −0.05, *p* = 0.78 | 0.08, *p* = 0.66 | −0.02, *p* = 0.91 | −0.15, *p* = 0.39 | −0.32, *p* = 0.06 |

[a] Sample size for all correlations *n* = 35; significant correlations at the 95% confidence level are highlighted in bold.

### 3.3. Predictors of DOM Indices: Linear Regression and RDA Models

GLM models identified significant predictors for DOM indices from a suite of hydrological or biogeochemical parameters (Table 2). Approximately 20–61% total variance was explained, except for %C4, for which no significant predictors were identified. The GLM models showed that water temperature was a significant positive predictor for SUVA$_{254}$ and %C1, but water temperature was a significant negative predictor for S$_R$ and %C3. Discharge was significant in predicting FI, BIX, and %C2 with positive coefficients, and $\delta^{18}$O was significant in predicting DOC and FI with negative coefficients. Sodium concentration was significant in negatively predicting HIX and positively predicting FI, BIX, and %C3; Si concentrations negatively predicted DOC but positively predicted S$_R$. Inorganic N and P nutrients (nitrate, ammonium, and phosphate) were not significant except that nitrate was selected for positively predicting SUVA$_{254}$.

**Table 2.** The generalized linear models predicting DOM indices in Mayfield Creek, the southeastern United States.

| DOM Indices | Generalized Linear Model [a] | R-Square |
|---|---|---|
| DOC | $-0.5486 \times$ **Si** $+ 0.3104 \times$ Nitrate $- 0.4945 \times \boldsymbol{\delta^{18}O} + 2.070\mathrm{e}^{-17}$ | 0.20 |
| SUVA$_{254}$ | $0.511 \times$ **T** $+ 0.1988 \times$ Si $+ 0.3608 \times$ **Nitrate** $+ 3.243\mathrm{e}^{-16}$ | 0.61 |
| S$_R$ | $-0.4901 \times$ **T** $+ 0.8060 \times$ **Si** $+ 0.4645 \times$ Q $- 3.675\mathrm{e}^{-16}$ | 0.39 |
| HIX | $0.2585 \times$ T $- 0.2211 \times$ Q $- 0.4335 \times$ **Na$^+$** $+ 1.930\mathrm{e}^{-16}$ | 0.40 |
| FI | $0.5681 \times$ **Q** $- 0.5096 \times \boldsymbol{\delta^{18}O} + 0.4041 \times$ **Na$^+$** $+ 5.934\mathrm{e}^{-16}$ | 0.32 |
| BIX | $0.4812 \times$ **Q** $- 0.2253 \times$ T $- 0.3443 \times$ d18O $+ 0.5079 \times$ **Na$^+$** $+ 1.79\mathrm{e}^{-16}$ | 0.51 |
| Percentage C1 | $0.4093 \times$ **T** $- 0.3043 \times$ Na$^+$ $- 1.184\mathrm{e}^{-16}$ | 0.25 |
| Percentage C2 | $0.3766 \times$ **Q** $+ 0.2392 \times$ T $+ 0.2567 \times$ Na$^+$ $- 1.808\mathrm{e}^{-16}$ | 0.25 |
| Percentage C3 | $-0.3650 \times$ **T** $+ 0.3537 \times$ Na$^+$ $- 1.387\mathrm{e}^{-16}$ | 0.25 |
| Percentage C4 | No reasonable model can be established from these predictors; the model with the lowest AIC has only intercept and R$^2$ = 0 | 0 |

[a] The model was selected based on acquiring the lowest AIC value. Significant parameters are highlighted in bold. T is water temperature measured in situ.

Results from the RDA model showed an overall consistent pattern with the GLM modeling results with regard to the relationships between DOM indices and environmental predictors (Figure 5). However, the RDA results also revealed a certain degree of sample grouping based on the season. Autumn samples were separated relatively well from other seasons, and only autumn samples fell in the third quadrant, where dissolved Si was positively aligned with HIX and %C4. Winter samples were also separated relatively well from other seasons, and the majority of winter samples fell within the fourth quadrant, where samples had high S$_R$ and %C3. Most of the summer and part of the spring samples were in the second quadrant, where nitrate and temperature were aligned positively with SUVA$_{254}$ and %C1. Finally, most spring samples, along with some samples from the other three seasons, fell in the first quadrant, where discharge, $\delta^{18}$O, sodium, phosphate, and ammonium were positively predicting DOC, %C2, BIX, and FI.

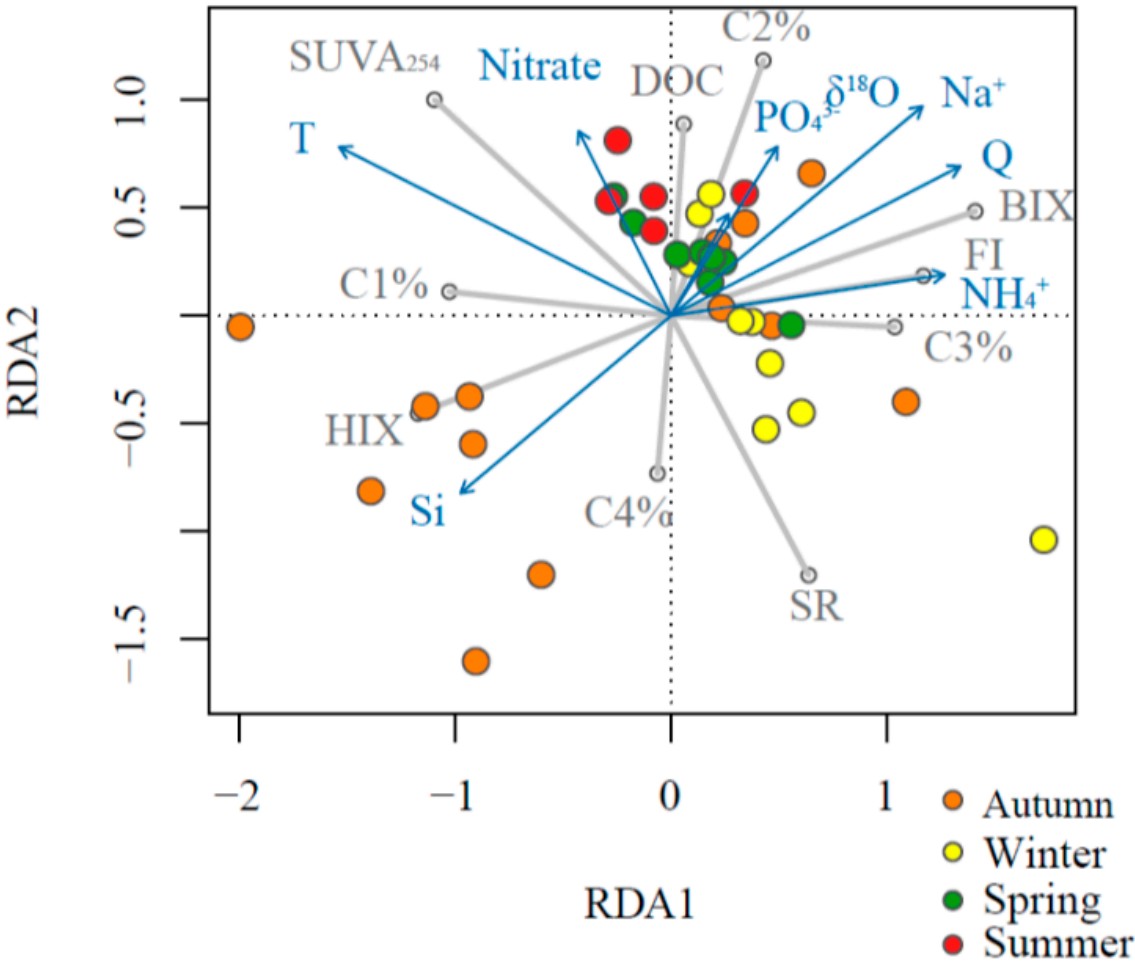

**Figure 5.** Redundancy analysis (RDA) plot of environmental variables as predictors for stream water DOM properties in the Mayfield Creek. Blue arrows denote predictors, grey lines denote dependent variables, and colored circles represent samples of different seasons.

### 3.4. DOM Yield

In Mayfield Creek, DOC concentrations ranged from 1.42 to 4.47 mg/L and averaged 2.36 ± 0.78 mg/L, and the instantaneous DOC flux (DOC concentration * discharge) varied from 0.11 to 1.64 g/s (0.64 ± 0.46 g/s). The DOC flux correlated more strongly with discharge ($r = 0.88$, $p < 0.001$, df = 33) than with DOC ($r = 0.56$, $p < 0.001$, df = 33). Similarly, the fluxes of four fluorescence components correlated more strongly with water discharge (Pearson $r = 0.50$–0.82, $p < 0.001$, df = 33) than with the intensity of fluorescence components (Pearson $r = 0.16$–0.51, $p =$ <0.001–0.36, df = 33). As a result, the DOC and DOM fluorescence yield followed the seasonal patterns of discharge (Figure 6). Overall, the lowest flux was observed during autumn, when water discharge was the lowest, and the highest flux appeared in winter with the highest water discharge. The spring and summer fluxes fell in the intermediate range, yet the summer flux was overall lower due to lower discharge.

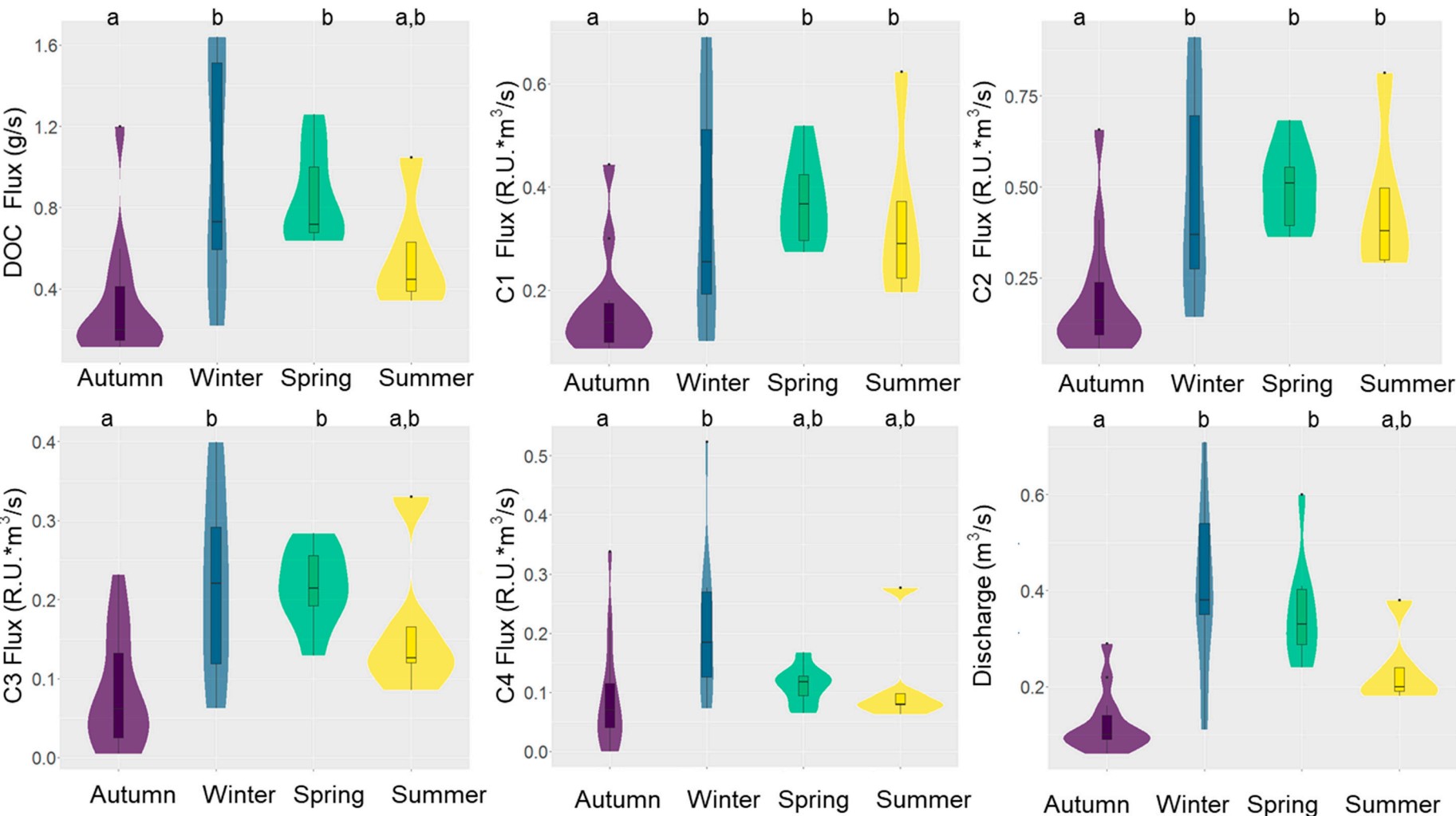

**Figure 6.** Violin and box plot comparison of DOC/DOM fluxes and discharge across seasons in the Mayfield Creek, southeastern United States. Boxes show the interquartile ranges (25th and 75th percentile), black lines across the boxes denote the median, and whiskers extend to 1.5 times the interquartile range. Black dots refer to outliers (i.e., beyond 1.5 times the interquartile range above the upper quartile and below the lower quartile). The width of color-shaded areas is proportional to the frequency of the corresponding Y values. Different letters above the boxes indicate significant differences detected by the non-parametric Kruskal–Wallis test with Dunn post hoc test.

## 4. Discussion

### 4.1. Source-Composition Characteristics of DOM Exported by Coast Plain Forested Streams

Four fluorescence components were identified in Mayfield Creek, and their relationships with DOM optical indices show different origins and reactivities of the four components (Table 3). Component 1 (C1) shows EEM characteristics consisting of a combination of traditional peaks 'A' and 'C' [38], and C1 is identified as humic materials exported from terrestrial sources that are usually of high molecular weights, high structural complexity, and great aromaticity [39,40]. Component 2 (C2) corresponds to the traditional 'M' peak [38] identified as humic substances that have been microbially processed and exhibiting relatively low molecular weights [41,42]. The positive correlation of %C2 and FI supports the microbial origin of C2 in Mayfield Creek, and the contribution of C2 is usually ascribed to soil microbes in small streams [25]. In Mayfield, C1 and C2 together accounted for nearly 70% of total fluorescence on average, suggesting the abundance of soil-derived DOM in the stream. However, it is important to note that the two humic DOM groups have different properties. Relative to terrestrial humic DOM (i.e., C1), microbial humic DOM (C2) is less humic and has lower aromaticity, as shown by the positive correlations of %C1 with $SUVA_{254}$ and HIX but the negative correlation between %C2 and HIX (Table 1). Additionally, terrestrial humic DOM is of diagenetic and recalcitrant nature, but microbial humic-DOM is more associated with recently produced DOM, as evidenced by BIX negatively correlating with %C1 but positively with %C2.

**Table 3.** Characteristics of the four fluorescence components identified by EEM-PRAFAC (DrEEM toolbox) and the attributed sources in Mayfield Creek, the southeastern United States.

| Component | Excitation Maximum Wavelength | Emission Maximum Wavelength | Similar Fluorescence Components Identified in Previous Studies | | | | Present Study |
|---|---|---|---|---|---|---|---|
| | | | Ref. [38] | Ref. [26] | Ref. [43] | Ref. [44] | |
| C1 | 250, 350 | 466 | A, C | C1 or SQ2 | C1 (Terrestrial) | C1 (Terrestrial Fulvic acid) | Terrestrial humic-like DOM |
| C2 | 250, 310 | 388 | M | C3 or Q3 | C4 (Microbial) | C3 (Microbial humic-like) | Microbial humic-like DOM from soils |
| C3 | 280 | 328 | B | C8 | C7 (Protein) | C4 (Protein-like) | Tyrosine-like, protein-like, autochthonous |
| C4 | <240, 290 | 352 | T | C13 | C8 (Protein) | - | Tryptophan-like, protein-like, autochthonous |

Two protein-like compounds were also identified in Mayfield Creek (Table 3). C3 is tyrosine-like and comparable to the 'B' peak, whereas C4 is tryptophan-like and comparable to the 'T' peak [38,45,46]. Components 3 and 4 are usually ascribed to autochthonous microbes and algae, and they together comprise a small fraction of fluorescent DOM (FDOM) (~30%), indicating the relatively low abundance of autochthonous DOM in the stream. However, our bioassay showed that the protein-like components were more reactive than the humic-like components (Figure 4), suggesting that protein-like DOM may play a more critical role in supporting in situ and downstream microbial metabolism. Previous findings also reported higher bioreactivity of protein-like fluorescence [7,47,48]. Although the two protein-like components are commonly assigned to the same sources, they displayed nuanced dissimilarities in Mayfield. The correlations between tyrosine-like DOM (i.e., C3) and DOM optical indices (Table 1) aligned with the expectation that protein-like DOM, relative to humic-like DOM, has overall lower molecular weights, aromaticity, and humification degree, but is more strongly associated with freshly produced, microbial DOM. Tryptophan-like DOM (C4), however, did not correlate with any DOM optical indices and could not be sufficiently explained by a GLM model (Table 2). The mechanism behind this observation is unclear, but it may be related to tryptophan-like DOM being more rapidly cycled and, thus, showing higher temporal variability. Previous studies have also noted dissimilar reactivities of tyrosine-like and tryptophan-like DOM, although the associated findings have been inconclusive. Some observed that tyrosine-like DOM was more bioreactive than tryptophan-like DOM [16,49], yet others have suggested that tryptophan-like fluorescence was more reactive to biodegradation and/or photodegradation [50,51].

The dominance of terrestrially derived humic FDOM in Mayfield Creek represents a typical characteristic of Coastal Plain streams draining a forest- or wetland-dominated landscape [52–54]. The Coastal Plain is one of the largest provinces in eastern North America, and headwater streams on the Coastal Plain represent important terrestrial–aquatic interfaces across which carbon and nutrients from the terrestrial landscape are mobilized to subsidize aquatic ecosystems. Many such streams are characterized as blackwater streams, named after their typical dark brown color due to high concentrations of dissolved organics from soils and decayed plants [55]. Blackwater streams and rivers are commonly associated with Coastal Plain landscapes with low relief. Mayfield has some features of blackwater systems, such as sandy benthic substrata with some woody debris and a tea color appearance of stream water (Supplementary Figure S1), but it has other characteristics that contrast with typical blackwater descriptions. Most notably, Mayfield has lower DOC concentrations (typically <5 mg C/L) than traditionally described for blackwater rivers (e.g., mostly >10 mg C/L), such as in the Satilla and lower Ogeechee rivers in the Coastal Plain of Georgia, U.S. [56]. Additionally, the Mayfield Creek watershed is forested in a gently hilly terrain in the upper Coastal Plain with only a few small wetlands formed from temporary beaver dams. It lacks the expansive floodplain swamps that generate the high concentrations of DOC observed in blackwater systems.

Although small percentages of humic substances can support measurable bacterial production (e.g., [57]), humic compounds (e.g., C1 and C2 in the present study) that dominate both in Mayfield Creek and other Coastal Plain streams are mostly biorefractory, as evidenced by our microbial degradation experiments (Figure 4) and many previous studies [47,48]. Proteinaceous compounds (e.g., C3 and C4) are more bioreactive than humic substances, being actively consumed and produced over the course of the incubations, but they account for a smaller fraction of DOM. Humic substances tend to be reactive to photodegradation [48], but forested Coastal Plain streams are usually well shaded by riparian plant canopies, which limits photodegradation. As such, these streams likely export much of the recalcitrant DOM, characterized by humic, aromatic-rich compounds, to downstream systems with minimal modification beforehand from microbial and photodegradative processes. Our data and interpretation agree with the general notion that much of OM added to small-sized streams (stream order 1–3) is degraded in mid-sized channels (order 4–6) in river networks [55]. Human modification, however, can significantly change the chemical characteristics of DOM and, as a result, shift the function of Coastal Plain streams. Previous studies [12,54] have linked human land use, including urban and agricultural lands, to increased bioavailability and reduced aromaticity and molecular weights of DOM in Coastal Plain streams, demonstrating that these streams can shift towards acting more as a biogeochemical reactor (as opposed to more like a transporter at natural state) due to human-induced enhancement in nutrient and light availability and alterations of carbon sources.

### 4.2. Temperature and Discharge Controls of DOM Source and Composition

Despite the relatively uniform source-compositional characteristics, DOM in Mayfield varies with several hydrological indicators. The GLM and RDA models assessed a suite of hydrological predictors, including stream water discharge (Q), water stable isotopes ($\delta^{18}$O and $\delta^2$H), Na$^+$, and dissolved Si. Water stable isotopes can measure the relative contributions of newer rainfall water vs. older subsurface water or groundwater [58]. Sodium concentration can indicate the hydrological connectivity between the stream with surface soils (i.e., relatively shallow flow path), whereas Si is an indicator of relatively deep flow paths or shallow groundwater contribution [25,59,60]. The models demonstrate an overall pattern that under high discharge with more new rainfall contributions and shallower flow paths, stream water DOC is higher, and DOM is more enriched in freshly-produced, low-humidity, microbial humic-like compounds. By comparison, low discharge and deep flow paths correspond to the addition of terrestrial humic-like, more-degraded compounds. This observation can be explained by that shallow flow paths during wet periods mobilize

freshly produced compounds from upper soil horizons, and deep flow paths during dry periods transport more recalcitrant, terrestrial humic compounds that have been degraded within the soil column. Moreover, high discharge decreases residence time of instream DOM and, hence, further reduces the effect of in-stream microbial and photochemical processing, although this reason may be secondary, as DOM from in-stream processing is relatively limited in Mayfield Creek. Our findings differ to some degree from the most commonly observed pattern from previous studies that DOM amount (DOC), aromaticity (SUVA$_{254}$), and terrestrial humic-like levels increase with rising discharge [18,61,62], since our data show that the aromaticity and terrestrial, humic-like DOM levels are more driven by temperature, as discussed below. This is likely because our samples were collected mostly during low-flow periods, when the flow path variation likely reflects water table changes and the associated loadings of near-stream riparian or hyporheic pool of OM [16], in contrast with those collected during hydrological events, where intense runoff can flush surface soil and leaf litter from more distant areas [9].

Our statistical models also identified the importance of temperature in influencing the preservation and input of recalcitrant DOM. Temperature is a significant predictor for SUVA$_{254}$, S$_{R}$, percentage of terrestrial humic materials, and percentage of tyrosine-like materials in Mayfield Creek, which indicates that high temperature favors the accumulation of aromatic, terrestrial, high-molecular-weight compounds but the removal of protein-like, low-molecular-weight compounds. This observation can be explained by that temperature mediated the diagenetic status of DOM via regulating microbial processing—higher temperatures promoted microbial removal (mineralization and uptake) of bioreactive, freshly produced compounds, leaving behind refractory compounds. In addition, temperature can influence root exudation and leaf litter degradation [63–65], and high temperature favors the releases of terrestrial, humic DOM from soils.

Inorganic nutrients, however, do not play an important part in regulating DOM source-composition characteristics (Table 2), substantiating the relatively minor importance of autochthonously derived organic matter on DOM quantity and quality in Mayfield Creek. Theoretically, nutrients can stimulate microbial production and processing of DOM and add autochthonous signatures. However, among the three nutrient predictors, only nitrate was selected as a significant predictor, and it positively correlates with DOM aromaticity (SUVA$_{254}$), indicating that autochthonous production does not drive this correlation. Rather, this correlation reflects that temperature, which positively influences SUVA$_{254}$ by promoting the preservation of recalcitrant compounds, also stimulates nitrate releases from soils and/or imparts a concentration effect on stream nutrients by increasing evapotranspiration. As Mayfield Creek is located in a federally protected national forest, other sources known to release nutrients, such as agriculture and urbanization, are not relevant.

The seasonal variations in DOM source-composition characteristics are collectively dictated by temperature and flow path (Figure 5). During the study period, winter had the highest overall discharge and lowest temperature, both of which favor the accumulation of freshly produced DOM. As a result, winter samples showed lower molecular weights and aromaticity but increased proportions of protein-like DOM. Autumn had the lowest discharge and relatively high temperatures, both of which favor the accumulation of recalcitrant, humic compounds. Deep flow paths leach more altered compounds from deeper soil horizons, and low discharge increases the time and effect of in-stream processing. In addition, litter fall peaks during the autumn, which can further contribute to the input of terrestrial, humic compounds. Summer DOM displayed greater aromaticity and higher molecular weights because of high temperatures that led to preferential preservation of recalcitrant compounds. Spring had relatively high discharge and thus showed higher DOC and greater percentage of microbial humic DOM, which indicates larger contributions of freshly produced DOM transported from upper soils under more reactive hydrological conditions. The flow and temperature variations in Mayfield Creek follow the general pattern in the southeastern U.S., where streamflow, evapotranspiration, and temperature variations are strongly seasonal, with periods of low flow occurring during summer and

autumn and periods of higher flow and flooding occurring during winter and spring [66]. As such, the seasonal DOM variations in Mayfield Creek are likely applicable to forested streams in the Southeast. However, two limitations of the present study need to be acknowledged. First, the duration of this study (slightly less than a year) was too short to capture year-to-year variations. Long-term monitoring practices, such as those being conducted by the National Ecological Observatory Network (NEON), are necessary to formulate a more robust understanding of longer-term temporal changes in these streams (note that Mayfield Creek is a NEON site, where long-term DOC, but not EEM-PARAFAC, is being collected along with other environmental parameters). Second, the instantaneous DOC concentrations and DOM fluxes were calculated from the values of low-flow periods, which are likely an underestimate of the amount of DOM exported from the stream, since these values typically rise during hydrological events [7,9,18].

Water discharge is the primary driver for the fluxes of DOC and all DOM fluorescence components (Figure 6), despite the seasonal variation in DOM source-compositional characteristics. As a result, greater quantities of all DOM components are exported in winter and spring than in autumn and summer. Given the short transit time and limited instream processing in these small, forested streams, the majority of DOM from these streams will be processed and utilized in locations further downstream, where conditions are favorable for microbial and photochemical alterations. Protein-like DOM is more bioreactive than humic-like DOM [7,47,48] and thus can contribute more significantly to stimulating microbial respiration and dissolved oxygen consumption in downgradient environments. Due to the high bioreactivity of proteinaceous compounds, they are likely to be rapidly consumed during downstream transport, yet extreme precipitation events and associated high flows can significantly accelerate the transport, as described by the pulse-shunt concept [67]. Labile DOM may be pulsed out of stream networks due to the high velocity of water and pushed far downstream to coastal water bodies, where bioreactive DOM can contribute to the creation of hypoxic "dead zones". On the other hand, the production and accumulation of humic DOM in the upstream is favored by high temperature, but the amount of humic DOM moving into downgradient networks depends more on water flux. Humic substances entering the water distribution system are a primary precursor of the formation of carcinogenic disinfection byproducts (DBPs), such as trihalomethanes (THMs) and haloacetic acids (HAAs), which are regulated substances in drinking water treatment [3,4]. As surface water bodies are a vital drinking water source in the southeastern U.S., high flow periods/seasons should be the priority when monitoring and managing natural DOM from the perspective of water quality.

## 5. Conclusions

Coastal Plain streams in southeastern U.S. supply carbon and nutrients to downstream systems and play an important role in the health of streams and downstream ecosystems because of their proximity to coastal oceans. Although numerous studies have been performed to understand hydroclimatic drivers for variations in DOM from small streams, few have focused on southeastern Coastal Plain streams. Here, we assessed hydrology and temperature controls of the amount, source-composition characteristics, and bioreactivity of DOM from a Coastal Plain stream. Our results show that FDOM is primarily controlled by terrestrial soil inputs, exhibiting relatively uniform properties of high aromaticity, large molecular weights, and the dominance of humic compounds. The seasonal variations in DOM are dictated by hydrology and temperature—high discharge and shallow flow paths correspond to fresher DOM with greater contributions of microbial humic-like and tyrosine-like compounds, whereas high temperatures favor the preservation of high-aromaticity, high-molecular-weight, terrestrial, humic-like substances. The total export of DOC and four fluorescence components are driven primarily by water discharge, and they are higher in winter and spring. Together, our results suggest that DOM biogeochemistry in forested, Coastal Plain streams does not involve instream processes that alter DOM in substantial ways before it is conveyed downstream. The quantity and composition of DOM exported

from these streams are closely regulated by temperature and discharge. This finding has important implications for future changes in biogeochemistry of Coastal Plain streams, as high temperatures and heavy precipitation are projected to become more prominent in southeastern U.S. in a rapidly changing climate.

**Supplementary Materials:** The following are available online at https://www.mdpi.com/article/10.3390/w13202919/s1, Figure S1: Mayfield Creek and in situ sensors, Figure S2: Rating curve in the Mayfield Creek, Table S1: Hydrological and nutrient parameters of the Mayfield Creek, Table S2: DOM parameters of the Mayfield Creek.

**Author Contributions:** Conceptualization, Y.L.; methodology, Y.L., P.S., Y.D. and M.B.; software, Y.L. and P.S.; validation, Y.L., P.S. and S.C.; formal analysis, Y.L., P.S.; investigation, Y.L. and P.S.; resources, Y.L.; data curation, Y.L. and S.C.; writing—original draft preparation, Y.L. and P.S.; writing—review and editing, Y.L., S.C., Y.D., M.B., A.K.W.; visualization, Y.L. and P.S.; supervision, Y.L.; project administration, Y.L.; funding acquisition, Y.L. All authors have read and agreed to the published version of the manuscript.

**Funding:** This research received no external funding.

**Data Availability Statement:** The data used in this study are provided in the Supplemental Materials.

**Acknowledgments:** The authors gratefully acknowledge use of the resources of the Alabama Water Institute at The University of Alabama. Y. Lu acknowledges the 6-month sabbatical Fellowship from Southern University of Science and Technology during the preparation of this manuscript.

**Conflicts of Interest:** The authors declare no conflict of interest.

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
