# Peer review of "Discharge and Temperature Controls of Dissolved Organic Matter (DOM) in a Forested Coastal Plain Stream"

_water, doi:10.3390/w13202919_

Round 1

Reviewer 1 Report

Generally say, it is good. The results are clearly presented and the scientific writing skill is great. I recommend getting published after some modifications. I have some suggestions:

Major suggestions:
In the introduction part, the authors may consider showing a big picture of significance. Or say, the author may consider using "scientific schematic illustration" to help the audience to understand the significance.

Minor suggestions:
(1) Line 137 Figure 2: Discharge Q. I cannot get the meaning of Q. Full-spelling is recommended in this figure.
(2) Line 218: Please check the font size.
(3) Line 297 Figure 4: 10/01/2015 has a small error bar in 5-day biodegradable fluorescence (%) C4. 11/29/2016 has a small error bar in 28-day biodegradable fluorescence (%) C4. The authors may check the data analysis. You may check this figure plotting and improve the visualization quality.
(4) Line 375 Table 3: Coble (2007); Cory and McKnight 2005; Yamashita et al. 2010; Lu et al. (2014). Please make it consistent before official publication.
(5) Line 542 Coastal Plain Stream and Line 556 Coastal Plain streams. Please make it consistent before publication.

Author Response

Please see the file attached. Thank you.

Reviewer 2 Report

Comments and Suggestions for Authors in attachment.

Author Response

Please see the file attached. Thank you!

Reviewer 3 Report

Overall this is a fairly well designed routine study investigating the composition of DOM and its biodegradability in a coastal plain stream in southeastern USA. The authors sampled within one watershed on a near-weekly basis for the better part of a year (September through July), analyzing DOM and stream water chemistry and biodegradability, building up a dataset to statistically model against physical environmental parameters. The authors found that DOM character was mostly dominated by flow patterns and was terrestrial in nature, and not dependent upon inorganic nutrient presence.  

While the study seems adequately designed and laboratory and in general statistical analyses appropriate carried out, there are several comments below in terms of clarity of methods and presentation of data:

Figures: 

Fig 2 needs to be cleaned up - use same font, consistent colors, axis of 2A is cut off, y-axis spelling error in 2B. Why are line and bar graphs used on the same graph for different variables? It makes it near impossible to see them. Either split parameters up onto multiple graphs, or present everything as scatters or lines.

Fig 4 should be separated or moved to horizontal layout, it is hard to see where one graph ends and another begins. 5 and 28 day fluorescence should be on the same scale. Should denote in some way statistically different values for degradation.

Fig 5 - why is the label DOC in red? Not listed in the caption.

Comments on Text and Tables:

Lines 143-144 - should reference SI for sampling dates. Was only one location sampled? How is it known that single location is representative of the entire stream?

Lines 156-170 – how many total spectra were used to determine 4 components? Listed as footnote in Table 1, should be in method text for PARAFAC analysis. What validation was performed with the DrEEM toolbox? What software was used? Validation should be presented in the SI

Lines 186-189 - how is it known those sampling dates are representative of that season for evaluating the biodegradability of the DOM? What was the oxygen availability during incubation? Was O2 exhausted after 28 days?

Lines 213-233 - Are variables are not dimensionally homogenous (have different physical units), were they not normalized before analysis?

Lines 280-282 - Table 1 - The correlations with fluorescent optical parameters and fluorescent components largely seem to just correlate with the components that had fluorescence in that range. For example, HIX correlates with C1 and C3, but given the wavelengths where HIX is derived, this appears to just be an artifact of the definition of HIX. The relevance of this table and its interpretation should be revisited.

Lines 312-314 - Table 2 -The lack of a GLM to C4, is this relevant? There should be some comment as to the overall contribution of C4 to the fluorescent DOM pool? If temp data was acquired 34 km away from sampling site (re: caption in fig 2), is it relevant to include in the GLM?

Lines 363-364 – without knowing the quantum yields of C1 and C2 fluorescence or the other groups, it is difficult to say that accounting for 70% of fluorescence signal translates to a predominance of those two components, or of all the DOM in the stream. Fluorescence is not necessarily representative of the entire DOM pool, and if fluorescence quantum yields vary, not representative of relative concentrations of species either.

Lines 552-553 - This study did not look at in-stream processes – sorption to the stream bed, analyses of microbial community composition and variation of efficiency in DOM degradation along the path of stream flow, etc. and as such, this claim is overstated.

There are some font size difference throughout the text.

Author Response

(The authors gave the same response as above.)
